# Low-Dose Nicotine Activates EGFR Signaling via α5-nAChR and Promotes Lung Adenocarcinoma Progression

**DOI:** 10.3390/ijms21186829

**Published:** 2020-09-17

**Authors:** Mong-Lien Wang, Yi-Fan Hsu, Chih-Hsuan Liu, Ya-Ling Kuo, Yi-Chen Chen, Yi-Chen Yeh, Hsiang-Ling Ho, Yu-Chung Wu, Teh-Ying Chou, Cheng-Wen Wu

**Affiliations:** 1Department of Medical Research, Taipei Veterans General Hospital, Taipei 112201, Taiwan; monglien@gmail.com (M.-L.W.); cj093138@gmail.com (Y.-C.C.); 2Institute of Food Safety and Health Risk Assessment, National Yang Ming University, Taipei 112304, Taiwan; 3Institute of Clinical Medicine, National Yang Ming University, Taipei 112304, Taiwan; shen7187@hotmail.com; 4Institute of Biochemistry and Molecular Biology, National Yang Ming University, Taipei 112304, Taiwan; dearevan2000@gmail.com (Y.-F.H.); ireen1109@hotmail.com (Y.-L.K.); 5Division of Molecular Pathology, Department of Pathology and Laboratory Medicine, Taipei Veterans General Hospital, Taipei 112201, Taiwan; ycyeh2@vghtpe.gov.tw (Y.-C.Y.); hlho5@vghtpe.gov.tw (H.-L.H.); 6Division of Thoracic Surgery, Department of Surgery, Taipei Medical University Hospital, Taipei 110301, Taiwan; wuyc@vghtpe.gov.tw; 7Division of Thoracic Surgery, Department of Surgery, School of Medicine, College of Medicine, Taipei Medical University, Taipei 110301, Taiwan; 8National Health Research Institute, No. 35, Keyan Road, Zhunan Township, Miaoli 35053, Taiwan; 9Institute of Biomedical Science, Academia Sinica, Taipei 115201, Taiwan; 10National Chiao Tung University, No. 1001, University Road, Hsinchu 300, Taiwan

**Keywords:** Lung adenocarcinoma, nicotine, environmental smoke, α 5-nAChR, EGFR

## Abstract

Nicotine in tobacco smoke is considered carcinogenic in several malignancies including lung cancer. The high incidence of lung adenocarcinoma (LAC) in non-smokers, however, remains unexplained. Although LAC has long been less associated with smoking behavior based on previous epidemiological correlation studies, the effect of environmental smoke contributing to low-dose nicotine exposure in non-smoking population could be underestimated. Here we provide experimental evidence of how low-dose nicotine promotes LAC growth in vitro and in vivo. Screening of nicotinic acetylcholine receptor subunits in lung cancer cell lines demonstrated a particularly high expression level of nicotinic acetylcholine receptor subunit α5 (α 5-nAChR) in LAC cell lines. Clinical specimen analysis revealed up-regulation of α 5-nAChR in LAC tumor tissues compared to non-tumor counterparts. In LAC cell lines α 5-nAChR interacts with epidermal growth factor receptor (EGFR), positively regulates EGFR pathway, enhances the expression of epithelial-mesenchymal transition markers, and is essential for low-dose nicotine-induced EGFR phosphorylation. Functionally, low-dose nicotine requires α 5-nAChR to enhance cell migration, invasion, and proliferation. Knockdown of α 5-nAChR inhibits the xenograft tumor growth of LAC. Clinical analysis indicated that high level of tumor α 5-nAChR is correlated with poor survival rates of LAC patients, particularly in those expressing wild-type EGFR. Our data identified α 5-nAChR as an essential mediator for low-dose nicotine-dependent LAC progression possibly through signaling crosstalk with EGFR, supporting the involvement of environmental smoke in tumor progression in LAC patients.

## 1. Introduction

Lung cancer is one of the leading causes of cancer death worldwide, accounting for 29% of total cancer deaths in the US and over 1.4 million deaths worldwide, largely because most cases are diagnosed at late stages [1]. Lung cancer is classified into two major types: non-small-cell lung cancer (NSCLC, accounting for 85% of all lung cancer cases) and small-cell lung cancer (SCLC, 15% of all lung cancer cases); in which NSCLC is further subclassified into three major histological subtypes: squamous cell carcinoma (20–25%), adenocarcinoma (30–40%), and large cell carcinoma (15–20%) [2]. Tobacco smoking causes many lung cancers, among them, the majority are squamous cell carcinoma and small cell carcinoma. Adenocarcinoma, another histologic type of lung cancer with increasing incidence globally, has long been regarded unrelated to smoking since a considerable number of the lung adenocarcinoma (LAC) patients are non-smokers [3]. However, this argument overlooks the potential second hand smoking in non-smokers caused by environmental tobacco smoke, and therefore, may mislead people to underestimate the detrimental effect of smoking on LAC. Second hand smoking has become a global issue of public health; the damages second hand smoking can cause, such as increased cancer incidence, are supported by epidemiological and genomic research [4,5]. It recently develops as a policy issue whether to restrict the scope of smoking area to reduce the emission of environmental smoke. In Taiwan, 90% of female lung cancer patients have no smoking history, but the occurrence rate of adenocarcinoma in female lung cancer patients is as high as 80% [6]. Considering the high smoking prevalence (86%) among men, which, as expected, leads to a large amount of involuntary exposure of non-smoking women to environmental smoke [6], the effects of environmental smoke on LAC may be more substantial than we currently think. However, an experimental proof of second hand smoke on cancer progression is still missing.

Nicotine, a major component of tobacco, promotes cell proliferation and facilitates cancer genesis and progression [7,8]. Nicotine is absorbed rapidly across pulmonary epithelium, into the arterial circulation, traveling to the central nervous system [9,10]. Nicotine can reach a micromolar concentration in the blood of smokers, while inhalation of environmental tobacco smoke results in a nanomolar blood level of nicotine [11]. The half-life of nicotine in the body is approximately 2 h. Significant accumulation of nicotine and its inactive metabolite, cotinine, can be found in plasma, urine, and saliva after exposure to environmental tobacco smoke in a normal working day [11]. Most of the previous nicotine studies focused on micromolar level of nicotine, and hence, may be more relevant to smoking activity. The mitogenic effect of nicotine is mediated by nicotinic acetylcholine receptors (nAChRs) [12]. nAChRs are ligand-gated ion channels, which are mainly expressed in the plasma membranes of certain neurons and on the postsynaptic side of the neuromuscular junction; they are also expressed in some non-neuronal cells, including bronchial epithelial cells [13]. A functional nAChR is either a homo- or a hetero-pentamer; the composition and stoichiometry of the subunits (named α1-α10 and β1-β4) constituting the pentamer may have a profound impact on receptor pharmacology, cation selectivity, desensitization kinetics, and spatial distribution. The α5 subunit of nAChR (α5-nAChR) plays an important role in nicotine addiction. α5-nAChR gene variants altered nicotine responsiveness in cultured human cells [14,15]. The presence of α5-nAChR subunit, combined with α4 and β2 subunits, increased calcium conductance and enhanced nAChRs affinity to nicotine [16,17]. Gallego et al. have shown that overexpression of the cluster CHRNA5/A3/B4 increased nicotine-induced c-Fos expression in dopaminergic neurons [18]. In the transgenic mice overexpressing the human CHRNA5/A3/B4 cluster, increased sensitivity to the pharmacological effects of nicotine was observed compared to the control mice, expressing many α3β4-nAChRs in the brain [18]. Nicotine administration induced dose-dependent seizures in both genotypes, but were more marked in human CHRNA5/A3/B4 transgenic mice [18]. This may suggest that α5-nAChR in mice increased cellular sensitivity to nicotine and modifies its reinforcing effects. Additionally, a study conducted by Gerzanich et al. indicated that in some types of nAChR, the α5-nAChR subunit increased Ca^2+^ permeability and specifically increased acetylcholine sensitivity of the α3β2 nAChR [19]. Moreover, the transcriptional deregulation at the *15q25* locus, which encodes for α5-nAChR, was associated with LAC risk [20,21,22,23]. Although the correlation of α5-nAChR with lung cancer progression is evident, the mechanism by which it regulates LAC tumor progression, especially in response to low-dose nicotine remains unclear [24].

In the present study, we evaluated the effect and the role of α5-nAChR in low-dose nicotine induced lung adenocarcinoma progression. We examined the α5-nAChR levels in LAC patients, as well as lung cancer cell lines, to establish the correlation between α5-nAChR expression and LAC. We investigated the functional and physical interaction between α5-nAChR and EGFR to demonstrate a crosstalk between nicotine and EGFR pathways. Our data revealed the essential role of α5-nAChR in regulating low-dose nicotine-dependent modulation of cancer cell motility, progression, and epithelial-mesenchymal transition (EMT).

## 2. Results

### 2.1. α5-nAChR Expression is Elevated in Lung Adenocarcinoma Cells and Clinical Tumor Tissue

We examined the expression of a panel of nAChR α subunit (*CHRNA*) genes in LAC cell lines and found that *CHRNA5* mRNA was the most expressed (Figure 1A). Further analysis in five LAC cell lines (A549, H1975, HCC827, CL1-0, and CL1-5) and two human bronchial epithelial cell lines (BEAS-2B and NL-20) demonstrated that *CHRNA5* mRNA levels were higher in LAC than in bronchial epithelial cells (Figure 1B). Comparing paired tumor and non-tumor tissues from 20 LAC and 10 squamous cell lung carcinoma (SCLC) surgical specimens (Figure 1C), we found higher levels of *CHRNA5* mRNA in tumor than in adjacent non-tumor tissues in LAC specimens; however, this differential expression pattern was not observed in SCLC samples. Since α7-nAChR and α9-nAChR have been implicated in the tumorigenesis of SCLC, gastric cancer, and breast cancer [25,26,27], we also examined their expression. Interestingly, *CHRNA7* particularly up-regulated in SCLC tumors, while *CHRNA9* did not show significant difference in tumor and non-tumor tissues in either LAC or SCLC. Immunohistochemistry results confirmed the overexpression of α5-nAChR protein in LAC tumor but not in the adjacent tissues (Figure 1D). Together, our findings suggest that α5-nAChR may be a specific type of nAChR subunit involved in LAC pathogenesis.

### 2.2. α5-nAChR Interacts with EGFR

Epidermal growth factor receptor (EGFR) is frequently overexpressed or activated in LAC, and promotes tumor initiation and progression in the early stages of cancer development [28,29]. Inhibition of EGFR signaling by EGFR tyrosine kinase inhibitor (TKI) has been an effective targeted therapy for advanced lung adenocarcinomas with EGFR-activating mutations in tumor cells. We established α5-nAChR-overexpressing or knockdown clones in LAC cell lines (Figure 2A) and found that α5-nAChR overexpression in LAC cell lines leads to enhanced EGFR phosphorylation as well as the phosphorylation of Akt and STAT3 (Figure 2B,C; Appendix AA). The levels of total EGFR were not changed much upon α5-nAChR overexpression and knockdown (Figure 2B,C); neither did the mRNA levels of EGFR (Appendix AB).

High-dose nicotine (0.5 μM) has been shown to activate the EGFR pathway in breast cancer cells, although the underlying mechanism is not elucidated [30]. Treatment of LAC cells by different concentrations of nicotine showed that the EGFR phosphorylation can be triggered with nicotine at a low concentration between 10 to 100 nM (Figure 2D,E). However, this phenomenon was missing in α5-nAChR-knockdown A549 cells (Figure 2D). Treatments with low (100 nM) and high doses (1 μM) of nicotine revealed increased amount of phospho-EGFR and phospho-Akt in response to as low as 100 nM of nicotine in vector control cells, and overexpression of α5-nAChR enhanced while knockdown of α5-nAChR blunted the nicotine-induced phosphorylation of EGFR and AKT (Figure 2F). A time course treatment with 100 nM EGFR activated EGFR phosphorylation within 1 h and reached a peak level by 4 h; knockdown of α5-nAChR markedly suppressed this nicotine-induced EGFR phosphorylation (Figure 2G), indicating an essential role of α5-nAChR in the low-dose nicotine-elicited activation of EGFR.

Given the functional interaction between EGFR and α5-nAChR, we tested the potential binding between the two proteins. A co-immunoprecipitation assay indicated that HA-α5-nAChR could be co-precipitated with His-EGFR, and vice versa (Figure 3A,B). Endogenous co-immunoprecipitation of EGFR and α5-nAChR was also observed in A549 and H1975 cells (Figure 3C), and the association between EGFR and α5-nAChR could be enhanced by EGF and nicotine stimulation (Figure 3D–F). Taken together, these data indicated a physical interaction between EGFR and α5-nAChR, and this interaction can be enhanced by nicotine and EGF stimulation.

### 2.3. α5-nAChR is Essential for Low-Dose Nicotine-Regulated Cell Motility

The malignant transformation of tumor cells is frequently accompanied with enhanced cell motility and changes in cell morphology. Observing the α5-nACh-overexpression and knockdown cells under microscope, we noticed that, compared with control cells, α5-nAChR-overexpressing cells were smaller in size and displayed more spiky processes, while knockdown of α5-nAChR induced flat and enlarged cell morphology (Figure 4A; Appendix AC). The altered morphology was possibly a result of the transition between epithelial and mesenchymal phenotypes, because compared to the control cells, cells overexpressing α5-nAChR expressed lower levels of E-cadherin but higher levels of mesenchymal markers including Slug, Snai1, N-cadherin, and vimentin, while knockdown of α5-nAChR had an opposite effect (Figure 4B; Appendix AD). Moreover, α5-nAChR positively regulated cell migration and invasion in LAC cells (Figure 4C–E; Appendix AE–H).

It has been shown that high-dose (400 μM) nicotine increases cell motility through α7-nAChR in gastric cancer [27], while the effect of low-dose nicotine on cell motility has not been investigated. Using A549 and HCC827 stable clones, we demonstrated that 100 nM of nicotine was sufficient for increasing cell migration in in vitro wound healing, and knockdown of α5-nAChR eliminated the low-dose nicotine-induced migration (Figure 4F; Appendix AI). Low-dose nicotine caused an increase in Slug protein expression in LAC cells within 4 h, and knockdown of α5-nAChR reduced this effect (Figure 4G), suggesting that low-dose nicotine may regulate EMT through α5-nAChR. Given the interaction between α5-nAChR and EGFR, we investigated the role of EGFR in nicotine/α5-nAChR-mediated regulation of EMT and cell motility. Pre-inhibition of EGFR activity with gefitinib at their IC_50_ and IC_80_ (50 and 80% inhibitory concentrations) blocked the low-dose nicotine-dependent induction of phosphorylation of Akt and STAT3 or Slug protein expression (Figure 4H), and this phenomenon cannot be reversed by α5-nAChR-overexpression (Figure 4I). Furthermore, knockdown of EGFR expression inhibited cell migration despite the overexpression of α5-nAChR (Figure 4J). Taken together, these results suggest that α5-nAChR functions upstream of EGFR and that low-dose nicotine may activate this α5-nAChR/EGFR axis in LAC cells to mediate the regulation of Slug expression and cell motility.

### 2.4. α5-nAChR Promotes LAC Tumor Progression In Vitro and In Vivo

We next investigated the role of α5-nAChR in regulating LAC cell proliferation and tumor growth. Growth curve analysis in all the three LAC cell lines showed that adding 100 nM nicotine in the culture medium increased the growth rate of LAC cells, while knockdown of α5-nAChR expression decreased LAC cell growth and the growth-promoting effect of low-dose nicotine was no longer observed in the α5-nAChR-knockdown cells (Figure 5A); these results suggest that low-dose nicotine enhances proliferation of LAC cells through α5-nAChR. To investigate the effect of α5-nAChR in clonogenic growth, we performed colony formation assays under two-dimensional (2D) culture conditions with high or low percentages of serum supplementation. In the presence of 10% serum, control and α5-nAChR-knockdown HCC827 stable clones formed similar numbers of colonies; however, when serum concentration was reduced to 0.5%, while control cells still formed some colonies, α5-nAChR-knockdown cells were not able to form colonies (Figure 5B, top panel), indicating that α5-nAChR can help sustain the growth of LAC cells under the stress of serum starvation. Under the low-serum culture condition, adding 100 nM nicotine increased the number of colonies formed by control cells but had no effect on α5-nAChR-knockdown cells (Figure 5B, middle and bottom panels), suggesting that low-dose nicotine may enhance cell growth/survival through α5-nAChR under this condition. Subcutaneous transplantation of HCC827 stable clones into BALB/c-nu/nu mice demonstrated that tumor growth was markedly suppressed in mice transplanted with α5-nAChR-knockdown cells compared to that in mice transplanted with control cells (Figure 5C); compared with HCC827-shα5 tumors, the HCC827-SC tumors have a higher α5-nAChR expression (α5-nAChR) and proliferation index (Ki67) demonstrated by immunohistochemistry (Figure 5D).

### 2.5. α5-nAChR Expression Correlates with LAC Tumor Recurrence and Patient Survival

To explore the clinical significance of α5-nAChR, we performed immunohistochemistry on LAC tissue microarrays and analyzed the association between α5-nAChR expression (scored as in Figure 6A–D) and clinical outcome. In a cohort of 133 patients with surgically resected stage-I LAC, patients with higher α5-nAChR expression in tumor tended to have worse outcome in both overall (Figure 6E) and disease-free (Figure 6F) survival, although the apparent association between α5-nAChR levels and the overall survival did not achieve statistical significance (Figure 6E); high α5-nAChR expression was significantly associated with poorer disease-free survival (*p* = 0.006) (Figure 6F), suggesting that overexpression of α5-nAChR increases the risk of tumor recurrence in stage-I LAC patients. Since no post-operative adjuvant chemotherapy or radiation was given to patients in this cohort, considering our findings in cultured LAC cells, we speculate that in patients who had tumors with α5-nAChR overexpression, low-dose environmental nicotine may continue to facilitate the growth/survival of residual tumor cells after surgery, which contributes to early tumor recurrence and worse disease-free survival.

We also analyzed a cohort of 176 patients with variable stages of LAC, which included 74 patients with wild-type EGFR (EGFR-Wt) (42.0%), 52 patients with EGFR L858R mutation (29.5%), 47 patients with EGFR exon 19 deletion (26.7%), and three patients with EGFR L861Q mutation (1.5%). There appeared to be no overall survival difference between patients with high or low α5-nAChR expression tumors within the cohort (Figure 6G); however, upon triage according to EGFR mutation status, high α5-nAChR expression was significantly associated with reduced overall survival in patients with EGFR-Wt LAC (Figure 6H), but not in patients with EGFR-mutant LAC (Appendix AA). Analysis of the lung adenocarcinoma patients with wild-type EGFR in the TCGA database [31] also demonstrated a significant correlation between high expression levels of CHRNA5 mRNA and poor overall survival outcome of patients (Appendix AB). Given these results, it is conceivable that for tumors without EGFR mutations, environmental nicotine may promote the progression of tumors with higher α5-nAChR expression more than those with lower α5-nAChR expression, and therefore, the overall survival is worse in the former group. For tumors with EGFR mutations, tumor cells are addicted to active EGFR signaling, and therefore, environmental nicotine would play a minor role in promoting tumor progression, rendering no statistically significant differences in survival between the group with high α5-nAChR levels and the group with low α5-nAChR expression.

## 3. Discussion

In Asia, none-smokers constitute 30% of NSCLC patients [32], and are characterized by higher incidence of adenocarcinoma subtype, in comparison to smoker patients. However, non-smoking does not exclude the potential intake of nicotine that may be caused by second-hand smoking, which results in the intake of environmental low-dose nicotine. The effect of second-hand smoking on lung cancer remains unknown, as there is a lack of evidence on the molecular mechanism and a lack of pathological studies. The present study identified α5-nAChR as a regulator mediating the tumorigenesis effects of low-dose nicotine on lung adenocarcinoma in aspects of proliferation, migration, epithelial-mesenchymal transition, and EGFR pathway activation (Figure 6I). Our in vitro work has demonstrated how low-dose nicotine may promote tumor growth and metastasis. A suitable animal model representing the environmental nicotine uptake may need to develop to investigate the systematic effects of the low-dose nicotine/α5-nAChR signaling. We also provided clinical and animal evidence of α5-nAChR implication in tumor progression. Inhibition of α5-nAChR and the restriction of environmental smoking/smoke can be considered as potential novel approaches for managing lung adenocarcinoma and tumor recurrence prevention.

Since the genome-wide screenings done in 2008 for the associated nicotinic receptor with high risk of lung cancer [33,34,35], many studies have attempted to explore the effect of nicotine on lung cancer progress. Another nicotine receptor subunit, α7-nAChR, has been implied involving in nicotine-dependent tumor progression, including cell survival, chemotherapeutic drug resistance, cell migration, and stemness genes expression [27,36,37,38,39,40,41]. Smoking habit was related to the overexpression of α7-nAChR in small cell lung carcinoma and lung squamous cell carcinoma [42]. However, these studies have focused on the effects of high dose nicotine, defined as being on a micromolar level, which is closer to the simulation of nicotine intake through smoking. In the current report, we identified α5-nAChR as a mediator for the cellular response to low-dose nicotine treatment. We found that α5-nAChR was overexpressed in the clinical tumor samples from LAC patients, which is in line with a previous study [21]. We have also examined the expression levels of nAChR α7 and nAChR α9 subunits; neither of them was overexpressed in LAC tumor samples in comparison to their normal counter parts (Figure 1B). A previous study indicated that a heteropentamer of nicotinic receptor containing α5-nAChR subunit (α4β2α5) showed higher affinity to low-dose nicotine than the pentamer without α5 subunit [17]. We further showed that knockdown of α5-nAChR in LAC cells suppressed the low-dose nicotine-dependent enhancement of proliferation and survival; knockdown of α5-nAChR in mouse model dramatically inhibited xenograft tumor growth. A study conducted by Shulepko et al. showed that 100 nM of nicotine enhanced the expression of α7-nAChR and accelerated the proliferation of A549 cell line. In fact, our data also showed the up-regulation of α7-nAChR expression in LAC cell lines to a lesser extent than α5-nAChR (Figure 1A). We did not rule out the possibility that other nAChR subunits may also be involved in low-dose nicotine mediated tumor progression. However, notably, the α-Bungarotoxin used to inhibit α7-nAChR was shown in a study to suppress the 10 μM nicotine-induced expression of both α5-nAChR and α7-nAChR [43]. Furthermore, and interestingly, we found that α5-nAChR may also mediate the transformation in normal lung epithelial cells: overexpression of α5-nAChR in BEAS2B cells increased their in vitro tumorigenicity property in soft agar culture and caused hypertrophy in mice (Appendix A). A previous study indicated that high doses (10 μM) of nicotine can increase the expression of α5-nAChR in normal oral keratinocytes and induce its transformation [43], while we reported that low-dose (100 nM) nicotine can also lead to normal cell transformation. Though the involvement of the low dose nicotine/α5-nAChR pathway in tumor initiation of LAC still needs to be further investigated in a suitable animal model, our data suggest a positive role of α5-nAChR in tumor progression of LAC

Abnormally activation of EGFR, either mutation or amplification, is frequently identified in LAC patients, resulting in constitutively auto-phosphorylation of EGFR itself and activation of its downstream molecules, including STAT3 and AKT. Through nicotine has been reported by several groups to increase phosphorylation and activation of EGFR [30], how and through which subunit nicotine functionally interacts with the EGFR-dependent pathway is unclear. Our finding that α5-nAChR sustained cells to low serum conditions triggered our hypothesis that α5-nAChR may regulate growth factor-mediated survival pathway(s) crucial for LAC cells. We demonstrated that α5-nAChR increased EGFR phosphorylation at Y1068, as well as the downstream signaling of EGFR such as STAT3 phosphorylation at Y705 and Akt phosphorylation at S473, which has been linked to cell survival and migration [44]. Consistently, a recent report also demonstrated the involvement of α5-nAChR in the EMT process and tumor metastasis through regulating the Jab1/Csn5 signaling in lung cancer [45]. Another study revealed the role of α5-nAChR in regulating cell proliferation and migration in melanoma cells [46]. We further showed that α5-nAChR is essential of low-dose nicotine-induced EGFR phosphorylation as 100 nM nicotine had limited effect on phosphorylated EGFR when α5-nAChR was knocked down. A protein-protein interaction of between EGFR and α5-nAChR is demonstrated in this study, and we showed that this interaction is positively regulated by EGF and nicotine stimulation. We showed a potential cross-talk mechanism between EGFR oncogenic signaling pathway and nicotine-α5-nAChR-dependent pathway. Our data, along with recent published reports, revealed the involvement of α5-nAChR in tumor progression. We do not rule out the possibility that other nAChR subunits may interplay or be involved in the α5-nAChR/EGFR interaction. Owing to the fact that α5-nAChR needs to form a functional nicotinic receptor with other subunits, it is very possible that other subunits, such as α3, α4, β2, and β4, might also be involved in the crosstalk between α5-nAChR and EGFR. Further investigation is needed to address the involving molecules in this crosstalk.

Although the regulatory mechanism between α5-nAChR and EGFR is still unclear, previous studies have revealed potential mechanisms through which the nAChR ion channel can regulate the membrane-bound receptor signaling cascade such as the EGFR pathway. Most of the studies were done with the α7-nAChR possibly because of the homomeric nature of the α7-nAChR receptor, which makes the study relatively easier than the heteromeric nAChR receptors. Activated homomeric α7-nAChR results in an influx of Ca2+ into the cells, which trigger the release of epidermal growth factor (EGF). The increased release of EGF then autocrinally triggers the activation of the EGFR cascade [12]. Furthermore, α7nAChR regulates noradrenaline release in the brain [47] and the sympathicus of nerves [48] as well as some cancer cells [49]. These stress neurotransmitters activate β-adrenergic receptors to stimulate the release of EGF, leading to activated EGFR signaling cascade and the development and progression of numerous cancers [12,50,51]. Another case of the crosstalk between nicotinic acetylcholine receptor and cellular receptor signaling pathway is the α9-nAChR homo-pentamer receptor and Estrogen receptor in breast cancer cells. Previous studies have shown that α9-nAChR may activate PI3K downstream kinases such as AKT, PDK1, and RSK, which phosphorylate and trigger the nuclear localization of estrogen receptor that binds to the promoter of α9-nAChR to enhance its expression [25]. These previous finding may shed lights on how the α5-nAChR regulates the EGFR pathway. It should be noted that there is still a possibility that the α5-nAChR subunit may have a function other than as a subunit of the nicotinic acetylcholine receptor.

The effects of α5-nAChR on cell proliferation, migration, and EMT can be witnessed in both EGFR wild-type (A549) and EGFR mutant (HCC827 and H1975) cell lines (Figure 4 and Figure 5); the animal experiment using HCC827 also proved that α5-nAChR can affect the tumor growth in EGFR-mutant cells (Figure 5C,D). However, our clinical correlation study showed that the correlation between high levels of α5-nAChR and poor survival outcome of LAC patients did not reach statistically significance in the cohort of EGFR-mutant patients (Appendix A). EGFR target therapy is the standard first line treatment for lung cancer patients with EGFR mutations, and the data in Figure 4H showed that inhibition of EGFR activity by Gefitinib can block the effect of low-dose nicotine. Therefore, it is possible that the inconsistent results from EGFR-mutant patients may be due to the interference from EGFR TKI therapy in a certain population of these patients. More detailed studies may be needed to address this phenomenon.

In summary, the present studies identified α5-nAChR as a pro-oncogene mediating the tumor progression and tumorigenesis of LAC in vitro and in vivo. In our in vitro studies we showed that α5-nAChR mediates the effects of low dose nicotine in LAC, in terms of proliferation, motility, EMT, and EGFR pathway activation. α5-nAChR could be a potential marker for predicting the risk of LAC patients when exposed to environmental nicotine, as well as a potential target of cancer prevention to reduce the second-hand smoking-induced LAC. These reports also provide a clue to the high occurrence rate of LAC in the group of non-smoker patients.

## 4. Materials and Methods

### 4.1. Cell Lines and Culture Condition

Lung adenocarcinoma cell lines (including A549, H1975, and HCC827), human bronchial epithelial cell lines (including BEAS-2B and NL20), and HEK293T human embryonic kidney cells were obtained from the American Type Culture Collection and tested positive for human origin. The CL-0 and CL1-5 lung adenocarcinoma cell lines were established previously [52]. All lung cancer cells were grown in RPMI 1640 with 10% fetal bovine serum (Sigma-Aldrich, Burlington, MA, USA) in a 37 °C, 5% CO2 incubator. HEK293T cells were maintained in Dulbecco’s modified Eagle’s medium (Sigma-Aldrich, Burlington, MA, USA) supplemented with 10% fetal bovine serum (Hyclone, Marlborough, MA, USA) in a 37 °C, 5% CO2 incubator. BEAS-2B cells were grown in F12 medium (Sigma-Aldrich, Burlington, MA, USA) with 4% fetal bovine serum, in a 37 °C, 5% CO2 incubator.

### 4.2. Expression Plasmid and shRNA

The α5-nAChR overexpression plasmid was purchased from Thermo (MHS1010-7508410), which contains full length α5-nAChR cDNA. The shRNA against α5-nAChR (ID: TRCN0000061135; target sequence: CCTTCAGAACTGTTCCATGAA) and EGFR (ID: TRCN0000121068; target sequence: GCCACAAAGCAGTGAATTTAT) were obtained from the National RNAi Core Facility at the Genomic Research Center, Academia Sinica, supported by the National Core Facility Program for Biotechnology Grants of NSC (NSC 104-2319-B-001-002).

### 4.3. Western Blot

Western blots were performed as described [53]. Primary antibodies included anti-α5-nAChR (Abcam, Cambridge, UK), anti-EGFR, anti-pEGFR Y1086, anti-AKT, anti-pAKT S473, anti-STAT3, anti-pSTAT3 Y705 anti-Slug, anti-E-cadherin, anti-vimentin, anti-N-cadherin (Cell Signaling Technology, Danvers, MA, USA), anti-α-tubulin (Santa Cruz Biotechnology, Dallas, TX, USA), and anti-β-actin (Sigma-Aldrich, Burlington, MA, USA). HRP-conjugated secondary antibodies were obtained from Santa Cruz Biotechnology (1:5000).

### 4.4. Animals and Tumor Cell Transplantation

HCC827-SC and HCC827-shα5 cells were harvested, washed, resuspended in PBS, and mixed with an equal volume of Matrigel (BD Biosciences, San Jose, CA, UAS). Cells (in a total volume of 100 μL) were injected subcutaneously into the right dorsolateral side of the flank region of 8-week-old male BALB/c-nu/nu mice (BioLASCO, Taiwan). The tumors were measured with a caliper weekly up to 7 weeks; the mice were then sacrificed and the xenograft tumors were excised. All experiments were performed in accordance with the guidelines set forth by the European Communities Council Directive of November 24, 1986 (86/609/EEC). The study was approved by the institutional commission of Academia Sinica, Taiwan for the control of the maintenance and use of animals (protocol number: 13-103580 from 01-01-2014).

### 4.5. Clinical Specimens and Tissue Microarray Construction

Tumor tissues from patients with lung adenocarcinoma who underwent surgical resection were retrieved from the surgical pathology archives at Taipei Veterans General Hospital. The specimens were fixed in formalin and embedded in paraffin before being archived. The pathological stage was determined according to the 7th Edition Union for International Cancer Control/American Joint Committee on Cancer TNM classification. Hematoxylin and eosin-stained slides were reviewed by the pathologist, and representative areas of tumor tissue were selected for tissue microarray construction. Tissue microarrays were constructed with 3-mm cores of tumor tissue retrieved from the paraffin blocks for each case. In total, two sets of tissue microarrays were constructed. The first one was a stage I tissue microarray, which was composed of tumor tissues from 133 patients with stage I lung adenocarcinoma who underwent surgical resection between 1995 and 2000. The second one was a variable stage tissue microarray, which was composed of tumor tissues from 204 patients with variable stage lung adenocarcinoma who underwent surgical resection between 2002 and 2006. All procedures of tissues acquirements have followed the protocols as detailed in the Declaration of Helsinki and were reviewed by the Institutional Review Committee at Taipei Veterans General Hospital (ethical authorization number: 2016-03-005AC). The study was approved by the medical ethical committee of Taipei Veterans General Hospital, which waived the need for written informed consent from the patients because of the retrospective nature of the study and the emotional burden that would result from contacting the patients or their relatives to obtain consent.

### 4.6. Immunohistochemistry and Assessment of Staining

Five-micrometer-thick sections were cut from the tissue microarrays and stained with α5-nAChR antibody. The intensity of immunoreactivity was graded according the following method described previously [54]. Briefly, the intensity score was semi-quantitatively assessed according to the staining intensity of tumor cells, as follows: 0 (negative; Figure 6A), 1 (weakly positive; Figure 6B), 2 (moderately positive; Figure 6C), and 3 (strongly positive; Figure 6D). The percentage score was semi-quantitatively assessed according to the percentage of positive-stained cells as follows: 0 (0%), 1 (1–10%), 2 (11–50%), and 3 (51–100%). A composite staining index (SI) was derived from the product of the intensity score and the percentage score, giving values ranging from 0 to 9. α5-nAChR expression by immunohistochemistry was dichotomized into high and low categories based on the median value of SI.

### 4.7. Survival Analysis

Overall survival time was measured from the time of operation to death from any cause or to the date of last follow-up. Observations on patients alive at the last follow-up visit were censored. Time to recurrence was defined as the time from the date of operation to the date of disease recurrence. Patients who died of causes other than lung cancer or who were alive on the last follow-up date were censored. Survival curves were estimated using the Kaplan-Meier method, and survival differences between subgroups were compared using the log-rank test. A *p* value of < 0.05 was considered an indication of statistical significance. All analyses were carried out using SPSS 19.0 for Microsoft Windows (IBM Corporation, New York, NY, USA).

### 4.8. Statistical Analysis

The results are reported as mean ± SD. Statistical analysis was performed using Student’s t test as appropriate. Statistical analyses were done by SigmaPlot 13 software (Systat Software, San Jose, CA, USA). A *p* value < 0.05 was considered statistically significant.

## 5. Conclusions

The high incidence of lung adenocarcinoma (LAC) in non-smokers is a clinically unsolved paradox. The present study provides experimental evidence on how environmental low-dose nicotine promotes lung cancer progression. This study reveals the link between second hand smoking and lung adenocarcinoma, which has long been considered unrelated to nicotine intake, and supports the restriction of environmental smoking/smoke as therapeutic strategies for managing lung adenocarcinoma patients.

## Figures and Tables

**Figure 1 ijms-21-06829-f001:**
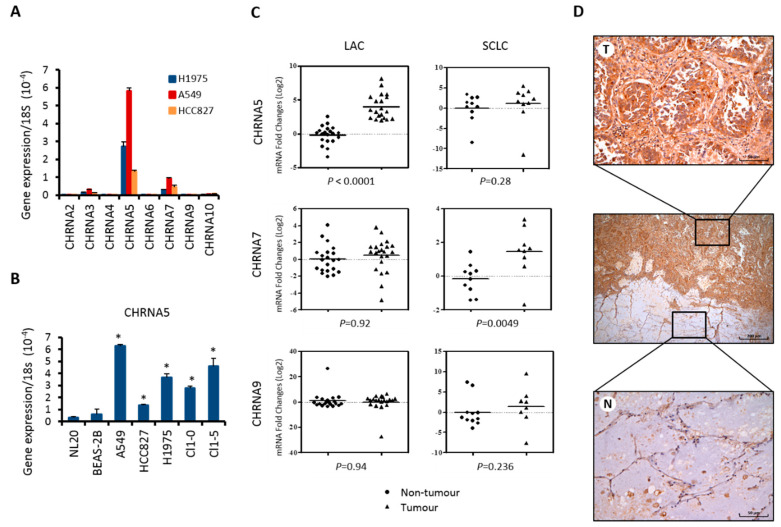
Expression of α5-nAChR was high in lung adenocarcinoma (LAC) cell lines and patient specimens. (**A**) LAC cell lines were analyzed by real-time quantitative reverse transcription PCR (qRT-PCR) to evaluate the mRNA expression of *CHRNA* genes encoding different alpha subunits of nAChR. Shown are relative expression levels normalized to the amount of 18*S* rRNA in each sample. Data represent the mean ± S.D. of three independent experiments performed in triplicate. (**B**) Human LAC (A549, HCC827, H1975, CL1-0, and CL1-5) and bronchial epithelial (NL-20 and BEAS-2B) cell lines were analyzed by qRT-PCR for mRNA expression of the gene encoding α5-nAChR (*CHRNA5*). Data represent the mean ± S.D. of three independent experiments performed in triplicate. Statistical analysis was performed using unpaired Student’s t test. * *p* < 0.05. (**C**) Twenty pairs of non-tumorous and tumorous LAC and 10 pairs of SCLC tissues were subjected to qRT-PCR analysis for *CHRNA5*, *CHRNA7*, and *CHRNA9* expression. Each symbol represents relative mRNA expression normalized to 18*S* rRNA in one sample. The line in each group represents the median value. Unpaired Student’s t test was applied to evaluate the significance. (**D**) Immunohistochemistry analysis of α5-nAChR protein expression in tumorous (T) and adjacent non-tumorous (N) lung tissues in LAC specimens. Original magnification, 40×; scale bar = 200 μm. High power views (original magnification, 200×; scale bar = 50 μm) are presented in the upper and lower panels.

**Figure 2 ijms-21-06829-f002:**
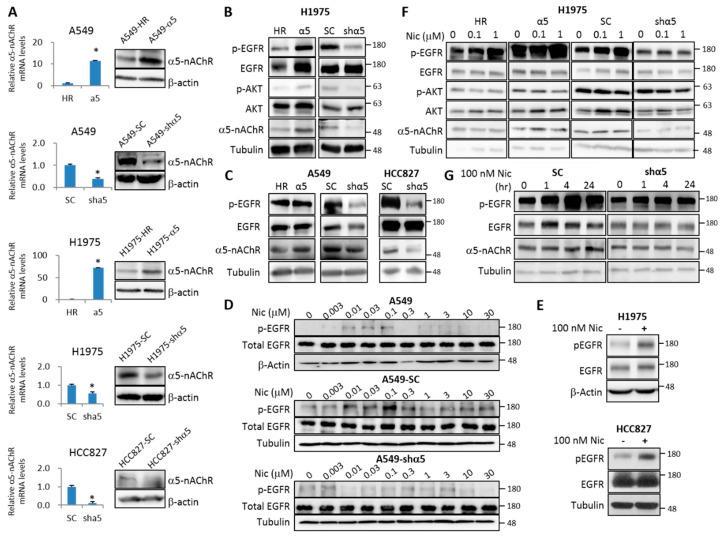
α5-nAChR mediates nicotine-induced activation of EGFR signaling. (**A**) α5-nAChR-overexpressing (α5) and knockdown (shα5) stable cells lines, and the respective empty vector (HR) and scrambled shRNA (SC) controls were established in A549, H1975, and HCC827 cell lines using a lentivirus transfection system. Cells were subjected to western blot analysis to assess the protein level of α5-nAChR. (**B**) H1975 stable cell lines were subjected to western blot analysis to assess the levels of indicated proteins. (**C**) A549 and HCC827 stable cells were analyzed by western blot to assess the expression level and phosphorylation status of indicated proteins. (**D**) A549 parental and stable cell lines were serum deprived for 24 h before 1 h treatment with variable concentration of nicotine. The phosphorylation status of EGFR was analyzed by western blot. (**E**) H1975 and HCC827 cells were serum deprived for 24 h before 1 h treatment with 100 nM of nicotine. The levels of total and phosphorylated EGFR were analyzed by western blot. (**F**) H1975 stable clones were serum-deprived for 24 h and subsequently treated with different doses of nicotine for another 24 h. Total cell lysates were subjected to western blot analysis. (**G**) H1975 stable clones were serum-deprived for 24 h before nicotine treatment. Total cell lysates were collected at 1, 4, and 24 h after the addition of 100 nM nicotine and analyzed by western blot. All data were repeated in at least three independent experiments to achieve statistical significance. Statistical analysis was performed using unpaired Student’s t test. * *p* < 0.05. The numbers at the right-hand side of the western blots indicates the sizes of the nearest protein markers.

**Figure 3 ijms-21-06829-f003:**
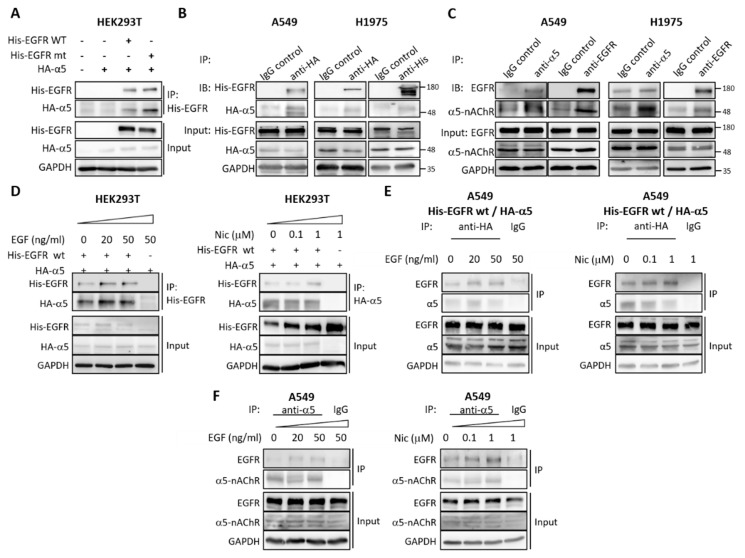
Protein-protein interaction between α5-nAChR and EGFR. (**A**) His-tagged wild-type and L858R mutated EGFR (His-EGFR WT and His-EGFR-mt) and HA-tagged nAChR α5 (HA-α5) expression plasmids were transfected in HEK293T cells, and the cell lysates were subjected to a co-immunoprecipitation assay with anti-His tag antibody to precipitate His-EGFR. The immunoprecipitated complexes were analyzed by western blot. (**B**) A549 and H1975 parental cells were transfected with His-EGFR WT and HA-α5, followed by immunoprecipitation with anti-HA or anti-His antibodies as described in (**A**). (**C**) Lysates of A549 and H1975 cells were subjected to immunoprecipitation with anti-α5 or anti-EGFR antibodies to pull down endogenous α5-nAChR or EGFR, and precipitated proteins were analyzed by western blotting. (**D**) HEK293T cells transfected with His-EGFR WT and HA-α5 were serum-starved for 8 h before being treated with EGF or nicotine for 1 h. Lysates were immunoprecipitated using anti-EGFR or anti-α5 antibodies, and precipitated proteins were analyzed by western blot. (**E**) A549 cells transfected with His-EGFR WT and HA-α5 were serum-starved for 8 h before being treated with EGF or nicotine for 1 h, followed by western blotting. (**F**) A549 cells were serum-starved for 8 h before being treated with EGF or nicotine for 1 h. Lysates were immunoprecipitated using anti-α5 antibodies, and precipitated proteins were analyzed by western blotting. All data were repeated in at least three independent experiments. The numbers at the right-hand side of the western blots indicates the sizes of the nearest protein markers.

**Figure 4 ijms-21-06829-f004:**
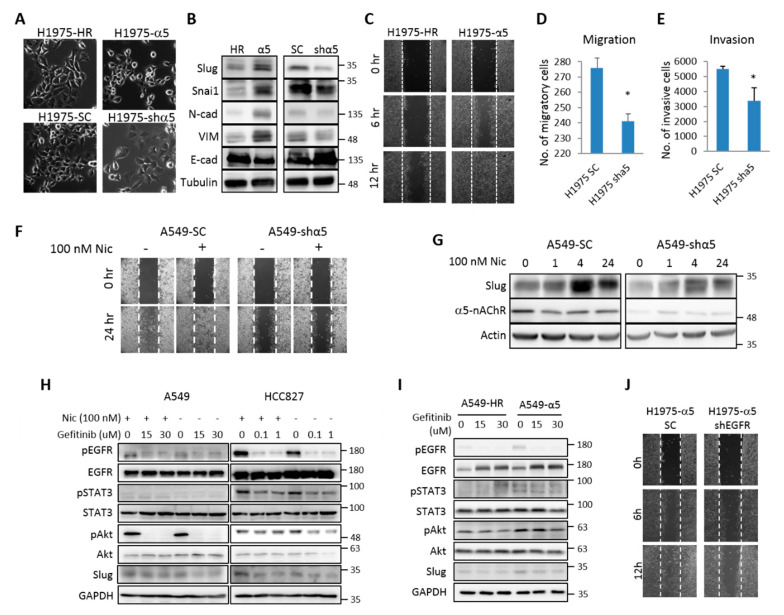
α5-nAChR is essential for low-dose nicotine-induced cell motility. (**A**). Micrographs showing the morphological changes induced by α5-nAChR overexpression or knockdown in H1975 cells. (**B**) H1975 stable cells were subjected to an analysis of EMT(epithelial-mesenchymal transition)-related protein expression by western blot. (**C**) Control (H1975-HR) and α5-nAChR-overexpressing (H1975-α5) cells were subjected to the wound-healing mobility assay for up to 12 h. (**D**,**E**) Control (H1975-SC) and α5-nAChR-knockdown (H1975-shα5) cells were subjected to Transwell assays for migration (**D**) and invasion (**E**) for 24 h. Numbers of migratory and invasive cells were quantified. Statistical analysis was performed using unpaired Student’s t test. * *p* < 0.05. (**F**) Control (A549-SC) and α5-nAChR-knockdown (A549-shα5) cells were subjected to a wound-healing mobility assay in the presence or absence of 100 nM nicotine for 24 h. (**G**) Control (A549-SC) and α5-nAChR-knockdown (A549-shα5) cells were serum-starved for 8 h, treated with 100 nM nicotine for indicated time, and analyzed by western blotting. (**H**) A549 and HCC827 cells were treated with Gefitinib at indicated concentrations 30 min before nicotine (100 nM) or vehicle treatment for another hour. Lysates were analyzed by western blotting for expression levels and phosphorylation status of indicated proteins. (**I**) A549 stable clones were treated with Gefitinib and subjected to western blot analysis as in (**H**). (**J**) EGFR-knockdown and control cells were established fromα5-nAChR-overexpressing (H1975-α5) stable clone. Cells were subjected to a wound healing migration assay for up to 12 h. All data were repeated in at least three independent experiments to achieve statistical significance. The numbers at the right-hand side of the western blots indicates the sizes of the nearest protein markers.

**Figure 5 ijms-21-06829-f005:**
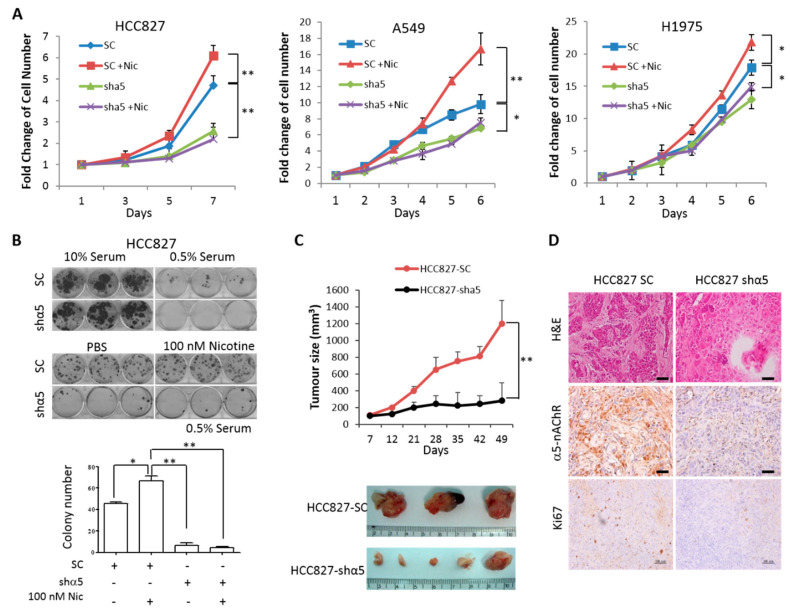
α5-nAChR positively regulates cell proliferation and tumor growth. (**A**) HCC827, A549, and H1975 stable cells were subjected to cell proliferation analysis by WST-1 assay in the presence or absence of 100 nM nicotine for 7 days. (**B**) HCC827-SC and HCC827-shα5 cells were seeded in 6-well plates at a density of 500 cells per well for colony formation assays. Cells were maintained in medium supplimented with either 10% or 0.5% serum (top) or in medium supplimented with 0.5% serum with or without 100 nM nicotine (middle) for 21 days before fixation crystal violet staining. The number of colonies in the middle panel were counted (bottom). (**C**) HCC827-SC and HCC827-shα5 cells were subcutaneously transplanted in 4-week-old female BALB/c-nu/nu mice (*n* = 5). The tumor size was measured by a caliper weekly and monitored up to 7 weeks. The xenograft tumors were excised and photographed 7 weeks after transplantation. (**D**) Immunohistochemistry staining for α5-nAChR expression in HCC827-SC and HCC827-shα5 tumors from mice in (**C**) Scale bar = 100 μm. All data were repeated in at least three independent experiments to achieve statistical significance. The quantification charts represent the mean ± S.D. of three independent experiments performed in triplicate. Statistical analysis was performed using unpaired Student’s t test. * *p* < 0.05, ** *p* < 0.01.

**Figure 6 ijms-21-06829-f006:**
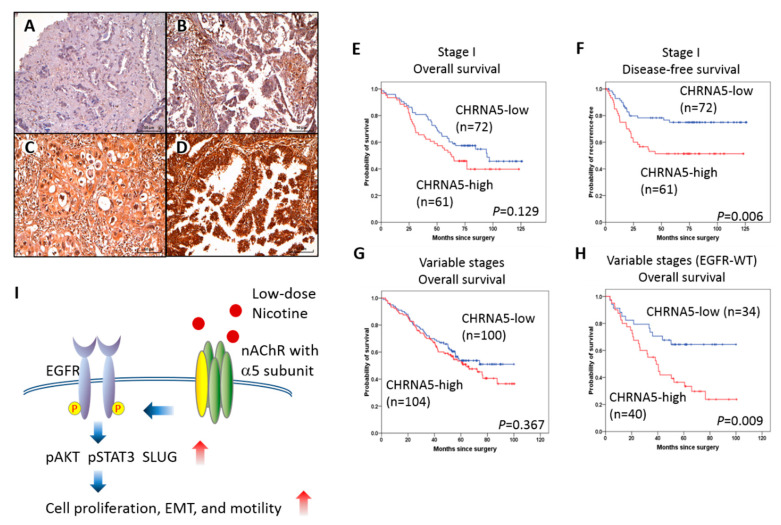
α5-nAChR positively associated with poor survival of EGFR-wild-type LAC patients. (**A**–**D**) Pathological scoring of α5-nAChR protein expression in LAC patients’ specimens: negative (d; intensity score = 0), weak expression (e; intensity score = 1), moderate expression (f; intensity score = 2), strong expression (g; intensity score = 3). Original magnification: 200×. Scale bar = 50 μm. (**E)** Kaplan-Meier analysis of the overall survival in stage I LAC patient cohort. (**F**) Kaplan-Meier analysis of postoperative disease recurrence in stage I LAC patient cohort. (**G**) Kaplan-Meier analysis of the overall survival in patients with EGFR-wild-type (WT) LAC at variable stages. (**H**) Kaplan-Meier analysis of the overall survival in patients with EGFR-mutant LAC at variable stages. (**I**) Schematic presentation of the α5-nAChR mediated tumorigenesis effects of low-dose nicotine on lung adenocarcinoma in aspects of EGFR pathway activation, cell proliferation, epithelial-mesenchymal transition, and cell motility.

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
