# Peer review of "Low-Dose Nicotine Activates EGFR Signaling via α5-nAChR and Promotes Lung Adenocarcinoma Progression"

_ijms, 2020, doi:10.3390/ijms21186829_

Round 1
Reviewer 1 Report
In the study "Low-dose Nicotine Activates EGFR Signaling via α5-nAChR and Promotes Lung Adenocarcinoma Progression", the authors demonstrate how the nicotine enhances lung adenocarcinoma and especially the cell proliferation, migration, and invasion; via EGFR and α5-nAChR pathway.
From a general point of view, the introduction gives a good overview of the context and the purpose of the study. The results part is well designed, organized, and clear. The discussion is interesting and well referenced. I also appreciated the concise conclusion.
From a more specific point of view, I have some comments regarding the results section:
- I would appreciate seeing scale bars on the immunohistochemistry pictures, to get a better appreciation of the observed structures
- I would also recommend adding the sample numbers on the figures or in the figures' legends. I am convinced that the authors used a consequent amount of samples and they should valorize their work by adding this information
- for the western blots, I would appreciate seeing the protein sizes on the figures
- in the figure 3D left panel, some information is missing to explain the purpose of the different wells
- I probably missed the information but would it be possible for the authors to precise to me where the figures 4D and 4E are mentioned in the text?
- In line 252, I guess the authors refer to Figure 5B instead of the 4B
Regarding the discussion part:
- I would suggest to the authors to delete the lines 317 to 319
- I am also wondering in this section if the authors refer to Figure 6I (e.g. line 327) or if the figure 7 is missing
For the materials and methods section:
- I would be glad to see the mention of the ethical authorization number for the human study
- for the immunohistochemistry part, again the Figure 7 is mentioned, so I have the same concerns as above
In conclusion, despite few minor adjustments, I appreciated reading this original work and wish all the best to the authors with this study.
Author Response
Comments and Suggestions for Authors
In the study "Low-dose Nicotine Activates EGFR Signaling via α5-nAChR and Promotes Lung Adenocarcinoma Progression", the authors demonstrate how the nicotine enhances lung adenocarcinoma and especially the cell proliferation, migration, and invasion; via EGFR and α5-nAChR pathway.
From a general point of view, the introduction gives a good overview of the context and the purpose of the study. The results part is well designed, organized, and clear. The discussion is interesting and well referenced. I also appreciated the concise conclusion.
From a more specific point of view, I have some comments regarding the results section:
- I would appreciate seeing scale bars on the immunohistochemistry pictures, to get a better appreciation of the observed structures
Thank you for indicating the missing scale bars. We have added scale bars in Figure 1D and Figure 6A-D. Please refer to the revised figures in the manuscript.
- I would also recommend adding the sample numbers on the figures or in the figures' legends. I am convinced that the authors used a consequent amount of samples and they should valorize their work by adding this information
We have added the sample numbers in figures legends to show the reasonable biological replicates of our data. Thanks for the reminder.
- for the western blots, I would appreciate seeing the protein sizes on the figures.
We have added the closest protein ladder markers to most of the Western blots. We also added the information about protein sizes in the list of antibodies in the new Supplementary Table S2. Please refer to the revised Supplementary Information.
- in the figure 3D left panel, some information is missing to explain the purpose of the different wells.
The information was mistakenly ignored when we pasted the figure in the manuscript. We have now replaced the figure with a new one with full information about each wells. We sincerely apologize for the mistake that should not have been happened.
- I probably missed the information but would it be possible for the authors to precise to me where the figures 4D and 4E are mentioned in the text?
The Figure 4D and 4E aim to delineate the effects of α5-nAChR in cell mobility (migration and invasion, using TransWell assay), which further confirms the claim in Figure 4C. Therefore, we added the citation of Figure 4D and 4E in line 220.
- In line 252, I guess the authors refer to Figure 5B instead of the 4B
Sorry for the type. We have corrected it into Figure 5B (line 267 and 270).
Regarding the discussion part:
- I would suggest to the authors to delete the lines 317 to 319.
Sorry for the mistake. We have deleted original lines 317 to 319 in the revised manuscript.
- I am also wondering in this section if the authors refer to Figure 6I (e.g. line 327) or if the figure 7 is missing
Yes. It was a mistake and we referred to Figure 6I. We have corrected it in the revised manuscript (line 347).
For the materials and methods section:
- I would be glad to see the mention of the ethical authorization number for the human study.
Thank you for the kind reminder. We have added the declaration and ethical authorization number (2016-03-005AC) for this study in section 4.5 in the revised manuscript.
- for the immunohistochemistry part, again the Figure 7 is mentioned, so I have the same concerns as above
We are sorry for the confusing description. It was a typo and has been corrected into Figure 6.
In conclusion, despite few minor adjustments, I appreciated reading this original work and wish all the best to the authors with this study.
We thank the reviewer’s constructive comments and suggestions, which largely improved our manuscript.
Submission Date: 09 July 2020
Date of this review: 30 Jul 2020 22:54:57
Reviewer 2 Report
This study reports that low levels of Nitocine at about 100nM is sufficient to activate/stabalize the cancer promoting effect of EGFR via the alpha5-nAChR receptor in lung cancer cells, which was based upon a large body of in vitro and in vivo experiments and corrobrative clinical data. In gereral, considering identification of the alpha5-nAChR is the receptor mediating the low-dose nitocine effect on EGFR, this finding is somewhat novel although activation of EGFR by nicotine is known. However, the data presented are largely correlative and not sufficient for supporting the current conclusion for the following reasons:
- The Western Blot data supporting alpha5-nAChR upregulates/stabilize EGFR are very weak, which are mostly likely due to variations of loading/imaging.
- Nicotin at 100nM was still able to significantly activate EGFR in cells whose nAChR has been knocked down by shRNA.
- Evidence of Nicotin activating the alpha5-nAChR receptor is missing.
- To test the role of alpha5-nAChR in mediating the effects of low-dose nicotine, knocking down alpha5-AChR is not approperiate due to the effects of potential unknown functions of the receptor and off-target effects of shRNA. Receptor antagonists or loss-of-function receptor should be employed to drawn such a conclusion.
- The in vivo test of the role of alpha5-AChR in mediating the cancer promoting effect of low-dose nicotine is deffective for missing treatment groups of nicotine and AChR antagonist.
- There is no data nor disscussion attempting to probe the mechanism of how the nicotine-alpha5-nAChR-EGFR activation works, in other words how does an ion channel activate EGFR?
- The central them of the study is to probe the role of low-dose nicotine in promoting lung cancers that expressing wild type EGFR, but oddly, the authors took the EGFR mutant HCC827 cells for in vivo testing, justification of choosing this model needs to be addressed.
Minor concerns:
1: The immunohistochemical data are not convincing due to OVER-staining, and the quantification method is not approperiate for normalizing the staining by area of tissue, becuase cells in cancer are more homogenousely cancers cells and cells in normal tissue are more heterogenouse. The staining intensity should be normalized at a level of single cells.
2: Figure 3D panel 1 missing treatment labels.
3: Figure 4H missing control treament of Nicotine alone
4: Figure 2D missing total EGFR.
Author Response
Comments and Suggestions for Authors
This study reports that low levels of Nitocine at about 100nM is sufficient to activate/stabalize the cancer promoting effect of EGFR via the alpha5-nAChR receptor in lung cancer cells, which was based upon a large body of in vitro and in vivo experiments and corrobrative clinical data. In gereral, considering identification of the alpha5-nAChR is the receptor mediating the low-dose nitocine effect on EGFR, this finding is somewhat novel although activation of EGFR by nicotine is known. However, the data presented are largely correlative and not sufficient for supporting the current conclusion for the following reasons:
- The Western Blot data supporting alpha5-nAChR upregulates/stabilize EGFR are very weak, which are mostly likely due to variations of loading/imaging.
We agree that the effect of alpha5-nAChR on total EGFR level is far weak than that on phosphorylated EGFR level. We modified the statement as follow: “The levels of total EGFR were not changed much upon α5-nAChR overexpression and knockdown (Fig. 2B-C); neither did the mRNA levels of EGFR (Fig. S1B)”. Please refer to the revised section 2.2 from line 150. We also quantified all the Western Blot data (listed in the revised Supplementary Figure S4) to avoid bias.
- Nicotin at 100nM was still able to significantly activate EGFR in cells whose nAChR has been knocked down by shRNA.
According to Figure 2F and 2G, the 100 nM Nicotine-mediated enhancement of EGFR phosphorylation in the α5-nAChR knockdown cells is far weak than the scrambled shRNA transfected controls. shRNA mediated knockdown may not remove the target gene expression completely, and often leave a certain percentage of the original expression level in cells. We believe that the left over α5-nAChR may still functional and affect the EGFR phosphorylation to a lesser extent.
- Evidence of Nicotin activating the alpha5-nAChR receptor is missing.
α5-nAChR is an auxiliary subunit that forms functional ion channels only when coexpressed with both α and β subunits[1]. It does not form a functional receptor on its own. And in cells, there are not only one type of nicotinic receptor expressing on the cell membrane. Therefore, it is not easy to specifically detect the activity of α5-nAChR receptor. In our study, we did not emphasize on the activity of the receptor, but to show that once this specific subunit (α5-nAChR) is overexpressed, cells react to low-dose nicotine-induced EMT and proliferation.
- To test the role of alpha5-nAChR in mediating the effects of low-dose nicotine, knocking down alpha5-AChR is not approperiate due to the effects of potential unknown functions of the receptor and off-target effects of shRNA. Receptor antagonists or loss-of-function receptor should be employed to drawn such a conclusion.
We thank the reviewer for the suggestion. Because the α5-nAChR is an auxiliary subunit that forms functional ion channels only when coexpressed with both α and β subunits[1]. It does not form a functional receptor on its own. So far there is no antagonists specifically against the α5-nAChR subunit. Using an antagonist against a broad group of the nicotinic receptor cannot identify the specific role of α5-nAChR. Moreover, the exact functional roles of this subunit is still unclear. Therefore, it is uncertain how to define a “loss-of-function” α5-nAChR. These are the reasons why we mainly used shRNA knockdown, accompanied with some data from CRISPR/Cas9 knockout, to dissect the role of α5-nAChR in this study. We understand our limitation at current status, and we keep working on the α5-nAChR downstream mechanism. We hope that we can answer some of the important questions the reviewer raised in the near future.
- The in vivo test of the role of alpha5-AChR in mediating the cancer promoting effect of low-dose nicotine is deffective for missing treatment groups of nicotine and AChR antagonist.
Thank you for the suggestion. The nicotine treatment in mice is under design for our next study that will dissect the downstream mechanism. We aim to use standard cigarette for mice inhalation in a defined cage. The animal experiments in this study is designed to prove that α5-nAChR is potentially a pro-tumor-promoting molecules that can enhance the tumor growth (Figure 5C-D) and even induce the transformation of normal cells (Supplementary Figure 3E-G). In these experiments, the α5-nAChR knockdown stable cells are compared with scrambled shRNA transfected cells whereas the α5-nAChR overexpression clones are compared with empty vector transfected clones. We believe that the experiments were logically designed with proper controls, and is adequate to support our claims.
- There is no data nor disscussion attempting to probe the mechanism of how the nicotine-alpha5-nAChR-EGFR activation works, in other words how does an ion channel activate EGFR?
We follow the reviewer’s suggestion and add a short discussion about the crosstalk between nAChR and EGFR networks, as listed below, in the discussion section. Please refer to the revised Discussion section in the updated manuscript from line 407.
“Although the regulatory mechanism between α5-nAChR and EGFR is still unclear, previous studies have reveal potential mechanisms through which the nAChR ion channel can regulate the membrane-bound receptor signaling cascade such as the EGFR pathway. Most of the studies were done with the a7-nAChR possibly because of the homomeric nature of the a7-nAChR receptor, which make the study relatively easy than the heteromeric nAChR receptors. Activated homomeric a7-nAChR results in the influx of Ca2+ into the cells, which trigger the release of epidermal growth factor (EGF). The increased release of EGF then autocrinally triggers the activation of the EGFR cascade [2]. Furthermore, a7nAChR regulates noradrenaline release in the brain [3] and the sympathicus of nerves [4] as well as some cancer cells [5]. These stress neurotransmitters activate β-adrenergic receptors to stimulate the release of EGF, leading to activated EGFR signaling cascade and the development and progression of numerous cancers [2,6,7]. Another case of the crosstalk between nicotinic acetylcholine receptor and cellular receptor signaling pathway is the α9-nAChR homo-pentamer receptor and Estrogen receptor in breast cancer cells. Previous studies have shown that the α9-nAChR may activate PI3K downstream kinases such as AKT, PDK1, and RSK, which phosphorylates and trigger the nuclear localization of estrogen receptor that binds to the promoter of α9-nAChR to enhance its expression [8]. These previous finding may shed lights on how the α5-nAChR regulates EGFR pathway. It should be noted that there is still a possibility that the α5-nAChR subunit may have alternative function other than a subunit of the nicotinic acetylcholine receptor. “
- The central them of the study is to probe the role of low-dose nicotine in promoting lung cancers that expressing wild type EGFR, but oddly, the authors took the EGFR mutant HCC827 cells for in vivo testing, justification of choosing this model needs to be addressed.
Thank you for your comment. In our in vitro studies, the effects of α5-nAChR (proliferation, migration, EMT) can be witnessed in both EGFR-wildtype (A549) and EGFR-mutant (HCC827 and H1975) cells. The animal experiment using HCC827 also proved that without other influence factors, α5-nAChR can also affect the tumorigenesis in EGFR-mutant cells. However, in our clinical correlation study, we showed that the correlation between high levels of α5-nAChR and poor survival outcome of patients did not reach statistically significance (Supplementary Figure S2). Because EGFR target therapy is the standard first line treatment for lung cancer patients with EGFR mutations, and from our data (Figure 4H) we know that Gefitinib can block the effect of low-dose nicotine. Therefore, we speculate that the inconsistent results from EGFR-mutant patients may due to the interference of EGFR TKI therapy in certain population of these patients. We have include this discussion in the revised manuscript. Please refer to the Discussion section of the updated manuscript from line 425.
Minor concerns:
- The immunohistochemical data are not convincing due to OVER-staining, and the quantification method is not approperiate for normalizing the staining by area of tissue, becuase cells in cancer are more homogenousely cancers cells and cells in normal tissue are more heterogenouse. The staining intensity should be normalized at a level of single cells.
As shown in Figure 6A-D, the α5-nAChR immunohistochemical stain demonstrated a wide range of protein expression from negative to intense strong expression. We believe the immunohistochemical staining is appropriate because it is able to capture the full range of α5-nAChR protein expression in lung adenocarcinoma tumor tissue in our cohort.
The quantification method in our study only includes tumor cells in the evaluation, and does not include normal tissue. Therefore, the quantification results will not be affected by the normal tissue present in the specimens.
- Figure 3D panel 1 missing treatment labels.
We are sorry for the mistake that should not have been happened. We have corrected this Figure with full label of each well. Please refer to the revised Figure 3D in the revised manuscript.
- Figure 4H missing control treament of Nicotine alone
The control of Nicotine alone treatment is the left lane of each group (100 nM Nic + / Gefitinib 0 mM).
- Figure 2D missing total EGFR.
Thank you for indicating this missing total EGFR. We have added total EGFR blots in the revised Figure 2D.
References:
- Arredondo, J.; Chernyavsky, A.I.; Jolkovsky, D.L.; Pinkerton, K.E.; Grando, S.A. Receptor-mediated tobacco toxicity: Acceleration of sequential expression of alpha5 and alpha7 nicotinic receptor subunits in oral keratinocytes exposed to cigarette smoke. FASEB journal : official publication of the Federation of American Societies for Experimental Biology 2008, 22, 1356-1368.
- Schuller, H.M. Is cancer triggered by altered signalling of nicotinic acetylcholine receptors? Nature reviews. Cancer 2009, 9, 195-205.
- Barik, J.; Wonnacott, S. Indirect modulation by alpha7 nicotinic acetylcholine receptors of noradrenaline release in rat hippocampal slices: Interaction with glutamate and gaba systems and effect of nicotine withdrawal. Molecular pharmacology 2006, 69, 618-628.
- Mozayan, M.; Lee, T.J. Statins prevent cholinesterase inhibitor blockade of sympathetic alpha7-nachr-mediated currents in rat superior cervical ganglion neurons. Am J Physiol Heart Circ Physiol 2007, 293, H1737-1744.
- Wong, H.P.; Yu, L.; Lam, E.K.; Tai, E.K.; Wu, W.K.; Cho, C.H. Nicotine promotes cell proliferation via alpha7-nicotinic acetylcholine receptor and catecholamine-synthesizing enzymes-mediated pathway in human colon adenocarcinoma ht-29 cells. Toxicology and applied pharmacology 2007, 221, 261-267.
- Schuller, H.M.; Tithof, P.K.; Williams, M.; Plummer, H., 3rd. The tobacco-specific carcinogen 4-(methylnitrosamino)-1-(3-pyridyl)-1-butanone is a beta-adrenergic agonist and stimulates DNA synthesis in lung adenocarcinoma via beta-adrenergic receptor-mediated release of arachidonic acid. Cancer research 1999, 59, 4510-4515.
- Laag, E.; Majidi, M.; Cekanova, M.; Masi, T.; Takahashi, T.; Schuller, H.M. Nnk activates erk1/2 and creb/atf-1 via beta-1-ar and egfr signaling in human lung adenocarcinoma and small airway epithelial cells. International journal of cancer 2006, 119, 1547-1552.
- Lee, C.H.; Chang, Y.C.; Chen, C.S.; Tu, S.H.; Wang, Y.J.; Chen, L.C.; Chang, Y.J.; Wei, P.L.; Chang, H.W.; Chang, C.H., et al. Crosstalk between nicotine and estrogen-induced estrogen receptor activation induces α9-nicotinic acetylcholine receptor expression in human breast cancer cells. Breast cancer research and treatment 2011, 129, 331-345.
Reviewer 3 Report
The study identifies α5-nAChR as a mediator of lung adenocarcinoma progression induced by low-dose nicotine consumption including that during passive smoking. The stimulatory role of α5-nAChR in lung carcinoma progression is shown in vitro and in vivo. As the formation of a functional network between nAChRs and EGFR is well-documented, the authors hypothesized interaction between α5-nAChR and EGFR and proved it by IP experiments. Some intracellular mechanisms of α5-nAChR effect on cell growth and motility, such as activation of EGFR and AKT as well as expression of mesenchymal markers are delineated by Western blotting. The implication of α5-nAChR in lung carcinoma progression is also confirmed by the clinical data.
In general, the research design is appropriate, and conclusions logically follow from the experimental results. Despite encouraging results, the following issues should be addressed to accept the article for publication:
Major points:
- Investigation of expression of CHRNA5, CHRNA7, and CHRNA9 in human lung carcinoma and non-tumor tissue samples (2.1), as well as the patient survival analysis (2.5), are impressive but that results can be further strengthened by analysis of mRNA expression in different databases, such as The Cancer Genome Atlas (TCGA). Brief Kaplan-Meyer analysis of TCGA lung adenocarcinoma data using Xena (https://xena.ucsc.edu/) showed that patients with high mRNA expression of CHRNA5 demonstrate significantly lower survival than patients with a low level of CHRNA gene, that completely fits the results of the current study. Moreover, as in the study, CHRNA5 the expression does not correlate with the survival of patients with EGFR mutated lung adenocarcinomas. You also can compare the expression of a different gene in lung adenocarcinoma and normal tissue (use “How I do…” section of Xena). It is not necessary to use Xena but TCGA database analysis should be included as a supplementary file for confirmation of your retrospective ICH analysis.
- Band intensities on western blots should be normalized to house-keeping genes, quantified and presented as supplementary files. This should be done at least for WB images in Fig 2, 3, and 4. For example, in Fig 2C (left panel, A549) enhancement of EGFR phosphorylation after CHRNA5 overexpression does not seem to be significant but as tubulin seems to be underloaded in CHRNA5 overexpressing cells, EGFR in CHRNA5+ cells can be actually activated. WB quantification can help to clarify those issues.
- As the most interesting results are obtained by Western blotting, the specificity of all antibodies used in the study should be demonstrated. This can be done by providing full-length membrane images with the molecular weight ladder as a supplementary image. For CHRNA5 and EGFR HEK293 cells may be used as the negative control (like in Fig 3A). Do not hesitate to show membranes with multiple bands as at least CHRNA5 has multiple sites of glycosylation. Also, please put the mol. weight of the protein bands at the right side of your WB frames on Figs.
- IP experiments are trustworthy but did you extract membrane fraction from cell lysate or precipitated proteins from total lysate? Clarify. Is it possible, that CHRNA5 and EGFR could precipitate because they were transported together to the cell membrane in the Golgi complex? Analysis of possible EGFR and CHRNA5 co-localization on the surface of lung adenocarcinoma cells by confocal microscopy can help resolve this issue.
- In 2.4, you can link the effects of low-dose nicotine on cell growth after serum deprivation with information about serum deprivation influence on CHRNA5 expression as well as on EGFR and AKT activation for H1975 cells. Information about serum deprivation effect on protein expression and phosphorylation in H1975 cells can be obtained after quantification of protein bands from Fig 2 – panel B for native cells, and panel F for serum-deprived cells.
- Why the overexpression of CHRNA5 was not shown for HCC827 cells? Consider the possibility of its addition.
- Comparison of CHRNA5 overexpression in normal and tumor cells is very valuable, so can you cite Fig S3 in the results section and discuss it? It was shown that CHRNA5 is overexpressed in normal keratinocytes after exposure to 10 µM nicotine [https://doi.org/10.1096/fj.07-9965.com], while you showed that low-dose nicotine can drive a normal cell’s transformation. Please discuss that issue. Also, discuss an article that also investigates low nicotine dose action on lung adenocarcinoma cells [https://doi.org/10.1016/j.intimp.2020.106303.]
- An ethical declaration concerning animal usage should be added in 4.3. Usually, it is formulated like “All experiments were performed in accordance with the guidelines set forth by the European Communities Council Directive of November 24, 1986 (86/609/EEC). The study was approved by the institutional commission of XXXX for the control of the maintenance and use of animals (protocol #XXXX from date-month-year).
- Please clarify the number of mice, used in the study. This should be done either in methods or in figure legends for Fig 5 and Fig S3.
- If the retrospective analysis of patient’s samples was approved by the Hospital’s Ethics Committee, the number and date of approval should be declared in 4.4. To deflect possible allegations of ethics violation you may declare that “The study was approved by the medical ethical committee of XXXX, which waived the need for written informed consent from the patients because of the retrospective nature of the study and the (emotional) burden that would result from contacting the patients or their relatives to obtain consent”.
- Was RNA treated by DNAse after isolation? Add that detail in Supplementary methods (qRT-PCR).
- Please provide information about primer design – if primers were picked up from literature provide a reference, if primers are designed by you, clarify in the methods section whether a primer pair is separated by at least one intron on the genomic DNA and provide amplicon length in Table S1.
- Catalog # and dilutions of all antibodies, including secondary, should be provided. Please note, that some Abs (Snai1, GPDH) are not mentioned in the Supplementary methods section. Also, I recommend putting the “Western blotting” section in the main text of the article as this is one of the core methods of investigation.
- In general due to big experimental work done the article is hard to follow. Please consider the usage of the English correction service.
Minor issues:
Abstract:
- α-nAChR should be replaced by α5-nAChR.
Introduction:
- Reference 10 does not seem to fit the author’s state. Please, check.
- Distribution of nicotine in the human body and possible metabolism should be described at least in one sentence.
- Reference 17 does not seem to contain notions about “cellular sensitivity to nicotine” it is better to say [line 84] “…increased nicotine-induced c-Fos expression in dopaminergic neurons”. Also, not only CHRNA5 but the cluster CHRNA5/A3/B4 was overexpressed in the transgenic mouse, please rephrase the sentence.
- Please decipher LAC abbreviation in the article text.
Results:
- Line 136 – there is no evidence of EGFR “positive regulation” by CHRNA5, please rephrase to …” that α5-nAChR overexpression in LAC cell lines leads to enhanced EGFR phosphorylation”.
- Could you justify why different proteins (actin, tubulin, or GPDH) were used as loading control?
- Line 191 – there is no “G” panel on the Fig. 3.
- Line 202 - clarify the transcription of protein Snail – if you meant “snail family transcriptional repressor 1” (Uniprot O95863) it usually is written as “SNAI” or “Snail1”, please provide one of those variants.
- Is there any possibility to measure the AKT and STAT3 phosphorylation by ICH of mice tumor transplants? If so, please add that data.
- Lines 256-257 – how tumor morphology can be more or less malignant? Please, either clarify or revise that state.
- Discussion:
- Lines 317-320 – text from the sample, please delete.
- nAChRs can form a functional complex with EGFR, is it possible that other nAChR subunits may interplay with CHRNA5 to form a complex with EGFR? Please, discuss this.
- Line 327 – there is no Fig. 7, maybe Fig. 6I
- Lines 327-330 – restriction of environmental smoking is rather “prophylactic” than actually “therapeutic strategy”. Also, inhibition of α5-nAChR EXPRESSION would not be performed easily in the clinic. My recommendation about that sentence: Our work has demonstrated how low-dose nicotine may promote tumor growth and metastasis and has also provided clinical evidence of α5-nAChR implication in tumor progression. Inhibition of α5-nAChR and the restriction of environmental smoking/smoke can be considered as potential novel approaches for managing lung adenocarcinoma and tumor recurrence prevention.
- Line 336 – article #39 was retracted, please use only #40 reference, it strongly supports your state.
Methods:
- Source of CL1-0 and CL1-5 cells should be added to 4.1.
- Clarify the software used for statistical analysis.
- Supplementary material:
- Please check the statistics in Fig S1B. It seems that EGFR expression in SC and sha5 A549 cells is significantly lower than that of HR and a5 clones. I recommend using a One-way ANOVA with a post hoc Tukey test for that situation.
- Supplementary methods, Proliferation assay -- …” cell density” can be replaced by “cell quantity”.
- Fig S3, please add scale to A panel.
The article requires some grammar corrections (as non-native English speaker I can not be sure about my suggestions):
- Line 44 – …” progression in LAC patients” instead…” progression of LAC patients”.
- Line 141 …“nicotine dose-course treatments” should be replaced by …“Treatment of LAC cells by different concentrations of nicotine showed that…”.
- Line 286 – avoid “and thus”.
- Line 324 – “study” instead of “studies”
- Line 371 -- …” involvement” instead of …” involvements”.
Author Response
Comments and Suggestions for Authors
The study identifies α5-nAChR as a mediator of lung adenocarcinoma progression induced by low-dose nicotine consumption including that during passive smoking. The stimulatory role of α5-nAChR in lung carcinoma progression is shown in vitro and in vivo. As the formation of a functional network between nAChRs and EGFR is well-documented, the authors hypothesized interaction between α5-nAChR and EGFR and proved it by IP experiments. Some intracellular mechanisms of α5-nAChR effect on cell growth and motility, such as activation of EGFR and AKT as well as expression of mesenchymal markers are delineated by Western blotting. The implication of α5-nAChR in lung carcinoma progression is also confirmed by the clinical data.
In general, the research design is appropriate, and conclusions logically follow from the experimental results. Despite encouraging results, the following issues should be addressed to accept the article for publication:
Major points:
- Investigation of expression of CHRNA5, CHRNA7, and CHRNA9 in human lung carcinoma and non-tumor tissue samples (2.1), as well as the patient survival analysis (2.5), are impressive but that results can be further strengthened by analysis of mRNA expression in different databases, such as The Cancer Genome Atlas (TCGA). Brief Kaplan-Meyer analysis of TCGA lung adenocarcinoma data using Xena (https://xena.ucsc.edu/) showed that patients with high mRNA expression of CHRNA5 demonstrate significantly lower survival than patients with a low level of CHRNA gene that completely fits the results of the current study. Moreover, as in the study, CHRNA5 the expression does not correlate with the survival of patients with EGFR mutated lung adenocarcinomas. You also can compare the expression of a different gene in lung adenocarcinoma and normal tissue (use “How I do…” section of Xena). It is not necessary to use Xena but TCGA database analysis should be included as a supplementary file for confirmation of your retrospective ICH analysis.
Thanks for the great suggestion. We have performed the survival analysis according to the CHRNA5 mRNA expression levels from the TCGA database. The data is included in the revised Supplementary Figure 2B. Please refer to the revised section 2.5 in the manuscript (description start from line 328) and Supplementary Information.
- Band intensities on western blots should be normalized to house-keeping genes, quantified and presented as supplementary files. This should be done at least for WB images in Fig 2, 3, and 4. For example, in Fig 2C (left panel, A549) enhancement of EGFR phosphorylation after CHRNA5 overexpression does not seem to be significant but as tubulin seems to be underloaded in CHRNA5 overexpressing cells, EGFR in CHRNA5+ cells can be actually activated. WB quantification can help to clarify those issues.
Many thanks for the suggestion. We have quantified all the WB in Figure 2, 3, and 4 and normalized each band intensity to the house-keeping genes. From the quantified results, the EGFR phosphorylation in CHRNA5 overexpressed A549 is more than that in the parental A549 cells in Fig. 2C left panel. The quantified results are now shown in Supplementary Figure S4 in the revised Supplementary Information.
- As the most interesting results are obtained by Western blotting, the specificity of all antibodies used in the study should be demonstrated. This can be done by providing full-length membrane images with the molecular weight ladder as a supplementary image. For CHRNA5 and EGFR HEK293 cells may be used as the negative control (like in Fig 3A). Do not hesitate to show membranes with multiple bands as at least CHRNA5 has multiple sites of glycosylation. Also, please put the mol. weight of the protein bands at the right side of your WB frames on Figs.
Thanks for the suggestions. We have added the closest protein ladder markers to most of the Western blots. We also added the information about protein sizes in the list of antibodies in the new Supplementary Table S2. Please refer to the revised Supplementary Information. The antibodies, apart from the anti-α5-nAChR, used in this study are all well-established and commonly used in many previously published reports. The anti-α5-nAChR is less used, and we showed the specificity with a full-length of WB as below.
- IP experiments are trustworthy but did you extract membrane fraction from cell lysate or precipitated proteins from total lysate? Clarify. Is it possible, that CHRNA5 and EGFR could precipitate because they were transported together to the cell membrane in the Golgi complex? Analysis of possible EGFR and CHRNA5 co-localization on the surface of lung adenocarcinoma cells by confocal microscopy can help resolve this issue.
The reviewer raised an interesting question. We used the total cell lysate for the IP experiments. Considering our large amount of IP data using different systems and cell lines, we believe that the protein-protein interaction between EGFR and α5-nAChR is well supported. The review’s question point out a question: where do these two proteins meet and bind. Following this study, we are actually working on the downstream mechanism after the protein-protein interaction. We found that dynamin-1 may also involve in this interaction, meaning that membrane structure and endocytosis status may be changed. We are working on that and wish to answer it in the near future. We thanks the reviewer for the excellent suggestion.
- In 2.4, you can link the effects of low-dose nicotine on cell growth after serum deprivation with information about serum deprivation influence on CHRNA5 expression as well as on EGFR and AKT activation for H1975 cells. Information about serum deprivation effect on protein expression and phosphorylation in H1975 cells can be obtained after quantification of protein bands from Fig 2 – panel B for native cells, and panel F for serum-deprived cells.
Thank you for the suggestion. We have quantified all Western Blots (including Figure 2B and 2F) and listed them in the revised Supplementary Figure S4.
- Why the overexpression of CHRNA5 was not shown for HCC827 cells? Consider the possibility of its addition.
Because for some reason, exogenous CHRNA5 is not well expressed in HCC827 cell. Therefore we only conducted knockdown studies in HCC827 cells.
- Comparison of CHRNA5 overexpression in normal and tumor cells is very valuable, so can you cite Fig S3 in the results section and discuss it? It was shown that CHRNA5 is overexpressed in normal keratinocytes after exposure to 10 µM nicotine [https://doi.org/10.1096/fj.07-9965.com], while you showed that low-dose nicotine can drive a normal cell’s transformation. Please discuss that issue. Also, discuss an article that also investigates low nicotine dose action on lung adenocarcinoma cells [https://doi.org/10.1016/j.intimp.2020.106303.]
Thank you for the suggestion. The Fig. S3 was mentioned in the Discussion section to discuss the role of CHRNA5 in inducing cell transformation in normal cell line. After the consideration of the readability and the flow of description, we think that to mention and discuss Fig. S3 in Discussion section would be easier for the audience to read without the interruption of the main focus on the role of CHRNA5 in cancer cells. We added the discussion including the mentioned study about oral keratinocyte in the revised Discussion. We thank the reviewer to improve our discussion with informative content. Please see the revised Discussion section from line 375.
The article published by Shulepko and colleagues showed that 100 nM of nicotine enhanced the expression of α7-nAChR and accelerated the proliferation of A549 cell line. But a comparative study on the effects of 100 nM nicotine on α7-nAChR and α5-nAChR was not conducted. The effects of specific knockdown of α7-nAChR in 100 nM Nicotine-induced cell proliferation was now investigated. In fact, the Fig. 1A in our manuscript also showed that α7-nAChR expression is elevated in LAC cell lines to a lesser extent than α5-nAChR. We did not rule out the possibility that other nAChR subunits may also involve in low-dose nicotine mediated tumorigenesis. However, it has to note that the α-Bungarotoxin used in this study to inhibit α7-nAChR was shown in the other study to suppress 10 nM nicotine-induced expression of both α5-nAChR and α7-nAChR[1]. We added a short description discuss this issue in the Discussion section. Please refer to the revised manuscript at line 368.
- An ethical declaration concerning animal usage should be added in 4.3. Usually, it is formulated like “All experiments were performed in accordance with the guidelines set forth by the European Communities Council Directive of November 24, 1986 (86/609/EEC). The study was approved by the institutional commission of XXXX for the control of the maintenance and use of animals (protocol #XXXX from date-month-year).
Thank you for the suggestion. We have added in the revised section 4.4 the declaration: “all experiments were performed in accordance with the guidelines set forth by the European Communities Council Directive of November 24, 1986 (86/609/EEC). The study was approved by the institutional commission of Academia Sinica, Taiwan for the control of the maintenance and use of animals (protocol number: 13-103580 from 01-01-2014)”.
- Please clarify the number of mice, used in the study. This should be done either in methods or in figure legends for Fig 5 and Fig S3.
Thank you for the reminder. In both in vivo experiments, 5 mice were applied for each experimental group. We have added the number of mice (n = 5) in the legends of the revised Figure 5C and Supplementary Figure 3G.
- If the retrospective analysis of patient’s samples was approved by the Hospital’s Ethics Committee, the number and date of approval should be declared in 4.4. To deflect possible allegations of ethics violation you may declare that “The study was approved by the medical ethical committee of XXXX, which waived the need for written informed consent from the patients because of the retrospective nature of the study and the (emotional) burden that would result from contacting the patients or their relatives to obtain consent”.
Thank you for the kind reminder. We have added the declaration and ethical authorization number (2016-03-005AC) for this study in section 4.5 in the revised manuscript.
- Was RNA treated by DNAse after isolation? Add that detail in Supplementary methods (qRT-PCR).
Yes, we treated the isolated total RNA with DNase before reverse transcribtion. We have added this information in the Supplementary Materials and Methods.
- Please provide information about primer design – if primers were picked up from literature provide a reference, if primers are designed by you, clarify in the methodssection whether a primer pair is separated by at least one intron on the genomic DNA and provide amplicon length in Table S1.
We have added these information about primer design as follow in the Methodology section in Supplementary Information. The amplicon length and location is also listed in the revised Supplementary Table S1.
“The primers were designed using the Vector NTI software through importing the coding sequences of target genes from NCBI database. The top ten primers were further analyzed by the open access Multiple Primer Analyzer software provided by ThermoFisher Scientific to avoid primer dimer. The picked primers were then blast with gene database in NCBK (pick primer blast) to confirm their specificity. The sequence of the primers designed to detect specific genes and the amplicon length are listed in Supplementary Table S1.”
- Catalog # and dilutions of all antibodies, including secondary, should be provided. Please note, that some Abs (Snai1, GPDH) are not mentioned in the Supplementary methods section. Also, I recommend putting the “Western blotting” section in the main text of the article as this is one of the core methods of investigation.
Thank you for these suggestions. We added additional supplementary table (Supplementary Table S2) to list all the antibodies, including 1st and 2nd antibodies, used in this study. The working dilution, suppliers, and catalog numbers are all listed in the table. The methodology of Western Blot is now placed in section 4.4 in the revised manuscript. Please refer to the revised Supplementary Information and revised Manuscript for these changes.
- In general due to big experimental work done the article is hard to follow. Please consider the usage of the English correction service.
Thank you for the suggestion. We have carefully revised the language of the manuscript and hopefully the readability is improved.
Minor issues:
Abstract:
- α-nAChR should be replaced by α5-nAChR.
Thank you for indicating the mistake. We have changed all the α-nAChR into α5-nAChR. Please refer to the revised Abstract in the manuscript.
Introduction:
- Reference 10 does not seem to fit the author’s state. Please, check.
Sorry for the mistake. We have removed the original reference 10, leaving reference 9, which is enough to support our statement in the introduction.
- Distribution of nicotine in the human body and possible metabolism should be described at least in one sentence.
We have added a few sentence in the introduction section.
“The nicotine is absorbed rapidly across pulmonary epithelium, into the arterial circulation, traveling to the central nervous system [2,3]. Nicotine can reach a micromolar concentration in the blood of smokers, while inhalation of environmental tobacco smoke results in a nanomolar blood level of nicotine [4]. The half-life of nicotine in the body is approximately 2 hours. Significant accumulation of nicotine and it inactive metabolite, cotinine, can be found in plasma, urine, and saliva after exposed to environmental Tabaco smoke in a normal working day [4]. “ Please refer to the revised Introduction in the manuscript (line 71).
- Reference 17 does not seem to contain notions about “cellular sensitivity to nicotine” itis better to say [line 84] “…increased nicotine-induced c-Fos expression in dopaminergic neurons”. Also, not only CHRNA5 but the cluster CHRNA5/A3/B4 was overexpressed in the transgenic mouse, please rephrase the sentence.
We thank the reviewer for the precise suggestion. We have modified the description as fallow: “Gallego et al. have showed that overexpression of the cluster CHRNA5/A3/B4 increased nicotine-induced c-Fos expression in dopaminergic neurons. In the transgenic mice overexpressing the human CHRNA5/A3/B4 cluster, increased sensitivity to the pharmacological effects of nicotine was observed comparing to the control mice majorly express a3b4-nAChRs in the brain. Nicotine administration induced dose-dependent seizures in both genotypes, but more marked in human CHRNA5/A3/B4 transgenic mice. This may suggest that α5-nAChR in mice increased cellular sensitivity to nicotine and modifies its reinforcing effects.” Please refer to the revised introduction section”. Please see the Introduction section from line 88.
In addition, we also including a reference from the study done by Gerzanich and colleagues, in which they indicated that in some types of nAChR, the α5-nAChR subunit increased Ca2+ permeability and specifically increased acetylcholine sensitivity of the α3β2 nAChR [5]. Please see the Introduction section from line 94.
- Please decipher LAC abbreviation in the article text.
We first deciphered this abbreviation in the abstract. Following the reviewer’s comment, we have added the full name of this abbreviation in the introduction section (line 58) where lung adenocarcinoma is first mentioned. This abbreviation is also listed in the “Abbreviations” section in the manuscript. Please refer to the Introduction an Abbreviations sections in the revised manuscript.
Results:
- Line 136 – there is no evidence of EGFR “positive regulation” by CHRNA5, please rephrase to …” that α5-nAChR overexpression in LAC cell lines leads to enhanced EGFR phosphorylation”.
Thanks for the comments. We have changed the sentence according to the reviewer’s suggestion for more precise description (line 149).
- Could you justify why different proteins (actin, tubulin, or GPDH) were used as loading control?
During a WB analysis we usually blot the membrane with several different antibodies, and even sometimes membrane stripping and re-blotting with another antibodie is required. Based on that, we will try to avoid overlap between target proteins and the house keeping genes. Therefore, different well-established loading controls were used in this study. They are all well-accepted and commonly used loading controls and we have tested that the WB using each of the house keeping gene as loading control did not divert the results.
- Line 191 – there is no “G” panel on the Fig. 3.
Sorry for the mistake. We have removed this mis-labelling in the revised manuscript.
- Line 202 - clarify the transcription of protein Snail – if you meant “snail family transcriptional repressor 1” (Uniprot O95863) it usually is written as “SNAI” or “Snail1”, please provide one of those variants.
It has been changed to Snai1. Please refer to the revised manuscript and Figure 4B.
- Is there any possibility to measure the AKT and STAT3 phosphorylation by ICH of mice tumor transplants? If so, please add that data.
Many thanks for the suggestion. The staining of phosphorylated AKT and STAT3 will visualize the activity of downstream signaling in vivo. However, the mice xenograft tumor samples did not preserved for phosphorylated protein and it is very possible that the phosphorylation signal has been lost during and handling and after the preservation. Staining with these samples may not reflect the real phosphorylation status of these proteins. Following this study, we are now working on the downstream signaling and the sub-cellular localization of several important molecules. We will validate the phosphorylation status and expression levels of our target molecules in vivo xenograft samples. We appreciate very much your kind suggestion which improves our study.
- Lines 256-257 – how tumor morphology can be more or less malignant? Please, either clarify or revise that state.
Thank you for the suggestion. We have revised the statement as follows: “compared with HCC827-shα5 tumor, the HCC827-SC tumor has higher α5-nAChR expression (α5-nAChR) and proliferation index (Ki67) demonstrated by immunohistochemistry (Fig. 5D).” Please refer to the revised manuscript from line 274.
Discussion:
- Lines 317-320 – text from the sample, please delete.
Sorry for the mistake. We have deleted original lines 317 to 320 in the revised manuscript.
- nAChRs can form a functional complex with EGFR, is it possible that other nAChR subunits may interplay with CHRNA5 to form a complex with EGFR? Please, discuss this.
We do not rule out the possibility that other nAChR subunits may interplay and involve in the α5-nAChR/EGFR complex. Actually, owing to the fact that α5-nAChR need to form a functional nicotinic receptor with other subunits, it is very possible that other subunits, such as α3, α4, b2, and b4, might also involve in the crosstalk between α5-nAChR and EGFR. We have included this statement in line 401 in the revised Discussion section.
- Line 327 – there is no Fig. 7, maybe Fig. 6I
Sorry for the mistake. We have corrected it into Fig. 6I
- Lines 327-330 – restriction of environmental smoking is rather “prophylactic” than actually “therapeutic strategy”. Also, inhibition of α5-nAChR EXPRESSION would not be performed easily in the clinic. My recommendation about that sentence: Our work has demonstrated how low-dose nicotine may promote tumor growth and metastasis and has also provided clinical evidence of α5-nAChR implication in tumor progression. Inhibition of α5-nAChR and the restriction of environmental smoking/smoke can be considered as potential novel approaches for managing lung adenocarcinoma and tumor recurrence prevention.
We appreciate the reviewer’s great suggestion on this description in Discussion section (line 347). We have modified the text according to the reviewer’s suggestion.
- Line 336 – article #39 was retracted, please use only #40 reference, it strongly supports your state.
Thank you for the reminder. We have removed the original reference No. 39.
Methods:
- Source of CL1-0 and CL1-5 cells should be added to 4.1.
Thank you for the suggestion. We added the source of CL1-0 and CL1-5 cell lines with reference in the revised section 4.1.
- Clarify the software used for statistical analysis.
The statistical analysis was done by SigmaPlot 13 software. This information has been included in the revised section 4.8.
Supplementary material:
- Please check the statistics in Fig S1B. It seems that EGFR expression in SC and sha5 A549 cells is significantly lower than that of HR and a5 clones. I recommend using a One-way ANOVA with a post hoc Tukey test for that situation.
Thank you for the suggestion. Following this suggestion, One-way ANOVA test did not show statistical significance of the slight changes on EGFR mRNA levels. This result correlates with our original claim that exogenous overexpression and knockdown of CHRAN5 affect EGFR phosphorylation and downstream activity but not EGFR mRNA levels in LAC cells.
- Supplementary methods, Proliferation assay -- …” cell density” can be replaced by “cell quantity”.
We have changed it according to the reviewer’s suggestion.
- Fig S3, please add scale to A panel.
The scale bars have been added to the panel A of revised Supplementary Figure S3.
The article requires some grammar corrections (as non-native English speaker I can not be sure about my suggestions):
- Line 44 – …” progression in LAC patients” instead…” progression of LAC patients”.
- Line 141 …“nicotine dose-course treatments” should be replaced by …“Treatment of LAC cells by different concentrations of nicotine showed that…”.
- Line 286 – avoid “and thus”.
- Line 324 – “study” instead of “studies”
- Line 371 -- …” involvement” instead of …” involvements”.
Thanks for the corrections. We have corrected them according to reviewer’s suggestions.
References:
- Arredondo, J.; Chernyavsky, A.I.; Jolkovsky, D.L.; Pinkerton, K.E.; Grando, S.A. Receptor-mediated tobacco toxicity: Acceleration of sequential expression of alpha5 and alpha7 nicotinic receptor subunits in oral keratinocytes exposed to cigarette smoke. FASEB journal : official publication of the Federation of American Societies for Experimental Biology 2008, 22, 1356-1368.
- Diaz, F.; Raval, A.P. Simultaneous nicotine and oral contraceptive exposure alters brain energy metabolism and exacerbates ischemic stroke injury in female rats. Journal of cerebral blood flow and metabolism : official journal of the International Society of Cerebral Blood Flow and Metabolism 2020, 271678x20925164.
- Lunell, E.; Bergström, M.; Antoni, G.; Långström, B.; Nordberg, A. Nicotine deposition and body distribution from a nicotine inhaler and a cigarette studied with positron emission tomography. Clinical pharmacology and therapeutics 1996, 59, 593-594.
- Jarvis, M.J.; Russell, M.A.; Feyerabend, C. Absorption of nicotine and carbon monoxide from passive smoking under natural conditions of exposure. Thorax 1983, 38, 829-833.
- Gerzanich, V.; Wang, F.; Kuryatov, A.; Lindstrom, J. Alpha 5 subunit alters desensitization, pharmacology, ca++ permeability and ca++ modulation of human neuronal alpha 3 nicotinic receptors. The Journal of pharmacology and experimental therapeutics 1998, 286, 311-320.
Round 2
Reviewer 2 Report
My major concern remains unresovled, which is that the in vivo experiments are insufficient to support the conclusion. Nicotine treatemnt groups must be included to draw the current conclusion.
Author Response
Thanks the reviewer for the critique. There is so far no standard and well-established animal models for nicotine uptake that mimicking smoking or secondhand smoking. Some used intravenous or intraperitoneal injection of nicotine in mouse while others added nicotine in water to feed the animals. However, none of these methods can represent the second-hand smoking or environmental nicotine uptake through the airway. We understand our limitation and, therefore, instead of emphasize the effect of nicotine, the major claim of our animal models is to support the roles ofα5-nAChR in tumor progression of lung cancer and in tumorigenesis of normal lung cells. The cell-based experiments presenting in Figure 2, 3, 4, and 5, as we believe, are able to support the reported effects of low-dose nicotine through α5-nAChR in LAC cell lines. The section title of our animal results is “α5-nAChR promotes LAC tumor progression in vitro and in vivo”, which clearly defined our focus on the oncogenic role of α5-nAChR in lung cancer. The animal experiment in supplementary information also states the role of α5-nAChR in cell motility and proliferation in BEAS-2B human bronchial epithelial cells. The oncogenic role of α5-nAChR was further supported by our clinical correlation data in which we showed that high level of α5-nAChR is correlated with poor survival outcome of LAC patients who harbor wild-type EGFR. We understand the limitation of current animal models and, therefore, we modified the first and last paragraphs of the discussion to specifically indicate our claims and also frankly mention the limitation of this study. Please refer to the following revised description in Discussion section.
“Our in vitro work has demonstrated how low-dose nicotine may promote tumor growth and metastasis. A suitable animal model representing the environmental nicotine uptake may need to develop to investigate the systematic effects of the low-dose nicotine/α5-nAChR signaling. We also provided clinical and animal evidence of α5-nAChR implication in tumor progression. Inhibition of α5-nAChR and the restriction of environmental smoking/smoke can be considered as potential novel approaches for managing lung adenocarcinoma and tumor recurrence prevention” from line 347.
“Though the involvement of the low dose nicotine/α5-nAChR pathway in tumor initiation of LAC still needs to be further investigated in suitable animal model, our data suggest a positive role of α5-nAChR in tumor progression of LAC” from line 382.
“In summary, the present studies identified α5-nAChR as a pro-oncogene mediating the tumor progression and tumorigenesis of LAC in vitro and in vivo. In our in vitro studies we showed that α5-nAChR mediates the effects of low dose nicotine in LAC, in terms of proliferation, motility, EMT, and EGFR pathway activation. α5-nAChR could be a potential marker for predicting the risk of LAC patients when exposed to environmental nicotine, as well as a potential target of cancer prevention to reduce the second-hand smoking-induced LAC. These reports also enlighten a clue to the highly occurrence rate of LAC in the group of non-smoker patients” from line 439.

Reviewer 3 Report
The authors have done a significant job of improving the article data presentation. They took into account most of the reviewer's comments and thoroughly argumented those lacking in the manuscript. The manuscript is suitable for publication, however, one minor point should be revised:
Line 403: ..."study to suppress the 10 nM nicotine-induced expression of both α5-nAChR"...
Within that article [https://doi.org/10.1096/fj.07-9965.com] effect of 10 μM (microM) but not 10 nM (nanoM) of nicotine on keratinocytes was investigated. Thus, [https://doi.org/10.1096/fj.07-9965.com] article shows that the high-dose nicotine can drive keratinocyte transformation, while you first showed that low-dose nicotine can also lead to the normal cell transformation. Please revise that issue.
Author Response
We thank the reviewer to recognize our efforts in revising the manuscript. We have changed the mistake and modified the discussion about the finding on the mentioned article as follow:
“However, it has to note that the α-Bungarotoxin used to inhibit α7-nAChR was shown in a study to suppress the 10 μM nicotine-induced expression of both α5-nAChR and α7-nAChR [43]” from line 375.
“A previous study indicated that high-dose (10 μM) nicotine can increase the expression of α5-nAChR in normal oral keratinocytes and induce its transformation [43], while we reported that low-dose (100 nM) nicotine can also lead to the normal cell transformation. Though the involvement of the low dose nicotine/α5-nAChR pathway in tumor initiation of LAC still needs to be further investigated in a suitable animal model, our data suggest a positive role of α5-nAChR in tumor progression of LAC“ from line 380.

Round 3
Reviewer 2 Report
The new discussion provides a fair justification for the conclusion. However the title of the paper remains unsupported by the results, which should be revised accordingly to the discussion. It may be changed to " Low-dose Nicotine Activates EGFR Signaling via α5-nAChR and α5-nAChR Promotes Lung Adenocarcinoma Progression"